# Assessment of Physicochemical Groundwater Quality and Hydrogeochemical Processes in an Area near a Municipal Landfill Site: A Case Study of the Toluca Valley

**DOI:** 10.3390/ijerph182111195

**Published:** 2021-10-25

**Authors:** Ingrid Dávalos-Peña, Rosa María Fuentes-Rivas, Reyna María Guadalupe Fonseca-Montes de Oca, José Alfredo Ramos-Leal, Janete Morán-Ramírez, Germán Martínez Alva

**Affiliations:** 1Geography Department, Autonomous University of the State of Mexico, Cerro de Coatepec s/n Ciudad Universitaria, Toluca 50110, Mexico; 2Engineering Department, Autonomous University of the State of Mexico, Cerro de Coatepec s/n Ciudad Universitaria, Toluca 50110, Mexico; 3Inter-American Institute of Technology and Water Sciences, Autonomous University of the State of Mexico, Carretera Toluca Atlacomulco Km 14.5, Unidad San Cayetano, Toluca 50200, Mexico; mgfonsecam@uaemex.mx; 4Applied Geosciences Division, Potosin Institute of Scientific and Technological Research, C.A. (IPICYT), Camino a la Presa San José # 2055, Lomas 4a, Sección, C.P., San Luis Potosi 78216, Mexico; jalfredo@ipicyt.edu.mx; 5Cátedras CONACYT, UNAM, Institute of Geophysics, National Autonomous University of Mexico, Ciudad Universitaria, Coyoacán, Cd. Mx, Mexico 04150, Mexico; janete@igeofisica.unam.mx; 6Medicine Department, Autonomous University of the State of Mexico, Tollocan Esquina Jesús Carranza S/N, Toluca 50180, Mexico; gmartinezal@uaemex.mx

**Keywords:** dissolved organic matter, leachate, anthropogenic contamination, aromatics proteins, groundwater quality

## Abstract

Sanitary landfills are considered one of the main sources of contamination of water resources due to the generation of leachate with a high content of dissolved organic matter (DOM), inorganic material, and toxic elements. This study aimed to determine the influence of leachate on the physicochemical quality and hydrogeochemical processes which determine the chemical composition of groundwater in an area near a municipal sanitary landfill site. In situ parameters (pH, temperature, electrical conductivity, dissolved oxygen, ORP), physicochemical parameters (HCO_3_^−^, PO_4_^3−^, Cl^−^, NO_3_^−^, SO_4_^2−^, NH_4_^+^, Ca^2+^, Mg^2+^, Na^+^, K^+^), and dissolved organic matter were analyzed. The content of dissolved organic matter (DOM) was determined by 3D fluorescence microscopy. The presence of Cl^−^, NO_3_^−^, NH_4_^+^, PO_4_^3−^, BOD, and COD indicated the presence of contamination. The significant correlation between NO_3_^−^ and PO_4_^3−^ ions (r = 0.940) and DOM of anthropogenic origin in the 3D fluorescence spectra confirm that its presence in the water is associated with the municipal landfill site in question. The type of water in the area is Mg-HCO_3_, with a tendency to Na-HCO_3_ and Na-SO_+_-Cl. The water-rock interaction process predominates in the chemical composition of water; however, significant correlations between Na^+^ and Ca^2+^ (r = 0.876), and between K^+^ and Mg^2+^ (r = 0.980) showed that an ion exchange process had taken place. Likewise, there is enrichment by HCO_3_^−^ and SO_4_^2−^ ions due to the mineralization of the organic matter from the leachate. The groundwater quality that supplies the study area is being affected by leachate infiltration from the sanitary landfill.

## 1. Introduction

The largest liquid freshwater resource on earth is stored as groundwater and is water for human consumption and agriculture. Groundwater is an essential and reliable source of water supply, despite climate change having generated significant spatial-temporal precipitation variability and affecting the storage volume in water reservoirs; however, occasionally pollutants reach the aquifer due to natural factors or human activities. One of the main threats to the quality of groundwater destined for human consumption is inadequate waste disposal sites, which affect all natural resources. This has a significant impact on the physical environment and human health. Landfills pose a severe threat to groundwater quality when they are operated incorrectly. The magnitude of this threat depends on the composition and quantity of leachate generated, the operation sanitary landfill time, and its distance from the aquifer. 

Groundwater contamination is of concern mainly during the sanitary landfill operation due to the pollutant load generated during solid waste degradation [1]. The leachate migration from landfills to groundwater is considered a serious environmental problem in uncontrolled and poorly designed municipal landfills [2]. The environmental impact caused by leakage of pollutants from the landfill is on the groundwater quality and is independent of selecting the ideal site for its location. The leachate from municipal landfills are effluents of complex composition with high concentrations, containing dissolved organic matter, inorganic compounds, heavy metals, and xenobiotic organic substances [3]. Therefore, it is vitally important to assess a potential risk associated with groundwater contamination from landfills.

Currently, the quality of groundwater is a critical environmental problem and is generally evaluated by monitoring dissolved organic matter (DOM) by measuring dissolved organic carbon [4,5,6,7,8,9]. DOM is a parameter that can reflect the contamination of water by organic compounds of anthropogenic origin. The organic matter of natural origin composed of humic and fulvic acids greatly influences the solubility and transport of harmful chemical contaminants. The high load of the latter can lead to the generation of carcinogenic by-products during disinfection processes [5,9,10]. The DOM present in the groundwater may differ from region to region, and the studies carried out to determine its origin report that it is predominantly derived from superficial garbage and soil, not from the aquifer matrix [5,9,11,12]. 

DOM is an essential component of the leachate generated in sanitary landfill sites and contributes more than 85% of the total organic matter in the organic carbon found in leachate. These leachates contain heavy metals and a significant quantity of DOM, with potential capacity in heavy metal speciation [10,12,13,14,15]. DOM composition varies depending on the composition of leachate, mainly in terms of the solid waste structure. It has been reported that there exists a strong affinity of humic substances (in DOM) with heavy metals due to the presence of a large number of functional groups in their structures that help form DOM-metal complexes [16,17]. Heavy metal contamination generated by leachate seepage from landfills has become increasingly worrisome due to its impact on human health as they can cause physiological effects [16,17,18]. Aquifers near landfills have a higher probability of contamination due to the incorporation of leachates into the water resource [19,20,21,22,23,24]. The deterioration of the quality of the groundwater near the sanitary landfill is a consequence of the low efficiency of its drainage systems [25]. The correct collection of leachates and the proper functioning of the landfill should ensure the quality of the groundwater below the landfill [26]. 

### Study Area

The area is located in the Municipality of Texcalyacac, in the State of Mexico. Texcalyacac is located at 19°09′15″ north latitude, 99°28′55″ west latitude, and the Greenwich meridian. It is a primarily urban community with 24.78 km^2^, 1799 hectares, with a population density of 208 per Km^2^. Texcalyacac accounts for 0.89% of the territory of the State of Mexico (Figure 1). Texcalyacac has a deep well (the source of the water supply for human consumption) and two re-pumping sites. The supply well in the study area is located 1.36 km from the source of water that feeds the Lerma River, considered one of the most polluted in the country. 

A sanitary landfill is a site assigned by a municipality to deposit its solid waste until it is time to transfer it to its final disposal. In Texcalyacac, there are three open-air landfills; one of them has exceeded its useful life, and none has leachate management and regulation of its solid waste. Therefore, these are temporary and do not comply with the characteristics assigned by Civil Protection; however, sanitary landfills continue to function without specific regulation and control, so since they do not have control, the leachate begins to infiltrate the aquifer until they find the groundwater and thus start the contamination process. (Figure 1).

Preliminary studies carried out in the study area frame the progressive alteration of the physicochemical quality of the water. The physicochemical analysis reports issued by these works show the presence of polluting species never seen before in the water that supplies the communities of the Toluca Valley. The polluting species found in groundwater are heavy metals such as arsenic, cadmium, cobalt, chromium, lead, fluoride ions, nitrogenous organic material, inorganic phosphate material, and aromatic proteins. Therefore, the present study aimed to evaluate groundwater quality and identify the hydrogeochemical processes in the aquifer near an unregulated municipal landfill [6,7,8,27].

## 2. Materials and Methods 

Five representative samples were taken over five months (March–November 2019) to cover the dry and rainy seasons. One of the samples was collected directly from the deep extraction well (~300 m) (S3), one from the Lerma River (S5), and three from the drinking water distribution points (S1, S2, and S4) in the Municipality of San Pedro Texcalyacac in the Toluca Valley. The collection of samples in the well was carried out before the chlorination process to preserve the original conditions of the groundwater. Three samples were taken at each sampling point: the first for physicochemical parameters, the second for elemental trace determination, and the third for 3D fluorescence analysis. Physicochemical parameters were determined according to standardized methods (APHA-AWWA-WPCF, 2005) [28]. Bicarbonates, chlorides, and hardness by volumetric method, biochemical oxygen demand by electrometric method, nitrates, ammonium and chemical oxygen demand by the TNT 835, TNT 831 and TNT 822 HACH technique respectively, phosphates by the stannous chloride method, sulfates by turbidimetric method, and chemical analysis of cations: calcium (Ca^2+^), sodium (Na^+^), magnesium (Mg^2+^), and potassium (K^+^), by the optic plasma inductive coupling (ICP) technique. The sample for 3D fluorescence analysis was filtered using nitrocellulose membranes (Ø 0.45 μm) and a vacuum filtration system [6,7]. For the fluorescence analysis method, the procedure proposed by Westerhoff et al., (2001) and Fuentes-Rivas et al., (2015) was used, and a Perkin Elmer Model LS55 Fluorescent Spectrometer was used [6,29]. The samples were transported to the laboratory and kept at 4 °C until analysis. The parameters of pH, water temperature (Tw), electrical conductivity (EC), total dissolved solids (TDS), and oxidation-reduction potential (ORP) were determined in situ with HANNA HI model 9146 equipment.

The statistical analysis (mean, maximum and minimum value) and Pearson’s correlation, to determine the dependence of the physicochemical parameters, was calculated with the Statistical Software Package for Social Sciences (SPSS). The diagrams to identify the type of water (Piper diagrams), the predominant hydrogeochemical process (Gibbs diagram), and the evolution of water (Mifflin and scatter diagram) were made with the Grapher11^®^ software. Figure 2 shows the flow chart of the methodology developed.

## 3. Results and Discussion

### 3.1. Physicochemical Parameters and Groundwater Quality

The data from the collected groundwater samples are summarized in Table 1. These include the following parameters: pH, Oxidation-Reduction Potential (ORP), Temperature (T), Electrical Conductivity (EC), Total Dissolved Solids (TDS), Hardness (Hard), Biochemical Oxygen Demand (BOD), Chemical Oxygen Demand (COD)), Dissolved Oxygen (DO), Phosphate (PO_4_^3−^), Chloride (Cl^−^), Ammonium (NH_4_^+^), Nitrate (NO_3_^−^), Sulfate (SO_4_^2−^), Bicarbonate (HCO_3_^−^), Sodium (Na^+^), Magnesium (Mg^2+^), Potassium (K^+^) and Calcium (Ca^2+^). Obtained results were compared to the WHO and Mexican standards for drinking water quality (NOM-127-Water for human use and consumption) [30,31].

The in situ values and most physicochemical parameters of all groundwater samples are within the WHO and Mexican standards range. Temperature measures the kinetic energy of water molecules and gives a value of the degree of heat that a substance possesses [27,28,29]. The temperature of the evaluated groundwater samples ranged between 20.66 °C and 24.15 °C (Table 1). The values of this parameter for all water samples were below the permissible limit established by the WHO for drinking water (25 °C) and complied with Mexican regulations [32], including the river water sample. A high water-temperature value can be attributed to environmental conditions and the climatic conditions that prevail in the study area when sampling. If variations in water temperature indicate a rising trend, this can give rise to chemical reactions between the aquatic environment and the geology of the site, causing the release of minerals from rocks. These minerals can represent a health risk when incorporated into the water in high concentrations; likewise, they lead to a decrease in levels of dissolved oxygen (DO) in the water [33]. As the temperature increases, the ability of water to hold dissolved oxygen decreases and, therefore, ORP levels drop. Thus, cold water can contain more dissolved oxygen than warm water, as happens in the analyzed samples. According to the correlation analysis, the temperature has a rather significant and negative correlation with magnesium (r = −0.601) (Table 2). 

Dissolved oxygen concentrations in samples ranged from 6.0 to 7.4 mg L^−1^. Samples 3 and 4 were within the range established by the WHO. Samples S1, S2, and S5 were slightly higher than the aforementioned limit. The highest concentration of DO was observed in S3 (second and fifth sampling). The concentration of oxygen present in the groundwater depends on the aquifer depth and the dynamics of the well; that is, a well with a high-speed extraction will allow the groundwater to mix with the interface water air-water, causing an increase in oxygen concentration [34,35]. Dissolved oxygen shows a positive correlation with ORP (r = 0.703) (Table 2), indicating the participation of oxygen in the oxidation of inorganic species.

The pH value of the groundwater in samples S1, S3, and S5 have levels of hydrogen ions which are within the permissible limit established by the WHO and Mexican regulations. However, in samples S2 and S4, all analyzed water samples have values above the safe limit (6.5–8.5) (Table 1). Therefore, the pH results show that samples two and four are outside the allowed limits in this study. Likewise, the lowest values were observed in the first sampling, caused by the dissolved salts present, as indicated by the correlation with the Cl^−^ (r = 0.516) and hardness (r = 0.612) (Table 2).

The chloride concentration in groundwater intended for human consumption is mainly due to anthropogenic factors or human activity. The concentration of chloride in the groundwater sample is directly related to the migration of wastewater into the aquifer. Likewise, it may be related to the inappropriate disposal of solid waste dumping in areas close to the water source [36,37]. The permissible limit for chloride in drinking water according to the WHO and Mexican regulations is 250 mg L^−1^ (Table 1). In the present study, chloride concentrations in water samples collected at all sites ranged from 6.06–53.75 mg L^−1^, all within the established limits (Table 1). A key factor in groundwater chloride accumulation is the porosity of the soil and the permeability of the rocks that exist in the area. However, the aquifer studied for the purposes of this paper is located in a volcanic environment, where fracture zones predominate, which facilitates the incorporation of this ion into the water, even though the aquifer reaches a depth of 300 m. The concentration of chlorides in the rainwater of the study area is approximately 1.0 mg·L^−1^, and in the geology of the site, there are no minerals that can contribute a significant amount of ion; therefore, its presence in the water may be associated with contamination by incorporation of leachate or discharge of residual water. 

Nitrogen is present in water mainly as nitrate (NO_3_^−^), but under reducing conditions, it can be present as ammonium (NH_4_^+^). Phosphorus is also present in water as a phosphate ion (PO_4_^3−^). Both nitrate and phosphate are an environmental concern as they are potential sources of nutrient enrichment in rivers, lakes, and wetlands. The presence of nitrates (0.0–3.17 mg L^−1^), ammonia (0–0.6 mg L^−1^), and phosphates (0.65–9.9 mg L^−1^) in groundwater are related to contamination by anthropogenic activities or infiltration of residual water [36,37,38], and this can cause variations in pH [39]. Agriculture is the primary source of nitrogen, while the origin of phosphate may be due to contamination by municipal wastewater, agriculture, anthropogenic activities, and the hydrological environment [40]. Table 2 shows a significant correlation between NO_3_^−^ and PO_4_^3−^(r = 0.940), and inverse correlation with SO_4_^2−^ (r = −0.776), which shows contamination by nitrogen and phosphate organic matter from residual water or leachate and not by fertilizers from the agricultural area.

The sulfate present in groundwater can be associated with mineral deposits in the form of sulfates in rocks, which tend to form oxides when they come into contact with water. In addition, there is the possibility of infiltration of industrial effluents with high concentrations of sulfates. According to the Mexican standard (NOM-127) and the WHO, the maximum permitted values of sulfates is 400 and 200 mg L^−1^, respectively. The data presented in Table 1 demonstrate that the sulfate values in the five analyzed sites ranged between 1.2 and 23.98 mg L^−1^. The N-NO_3_^−^ and N-NH_4_^+^ values for all sampling sites ranged between 0.2–3.2 mg L^−1^ and 0.0–0.6 mg L^−1^, respectively. Despite the low concentrations of chloride, sulfate, nitrate, and ammonium in groundwater, their presence is directly related to contamination by human activities. In this specific case, it is associated with the infiltration of leachates from the solid waste disposal site, which is located a few meters from the water source, and due to the contribution of fertilizers, due to its proximity to the agricultural area [41].

In addition to chlorides, ammonium and nitrate, two indications of contamination in groundwater are BOD (0–76.80 mg L^−1^) and COD (3.66–320.73 mg L^−1^). Neither of these parameters is regulated by Mexican or WHO standards. BOD shows a significant and positive correlation with N-NH_4_^+^ (r = 0.775). All three drinking water sources available in this area should be tested for bacteriological contamination, especially if the nitrate level is greater than 10 mg L^−1^, or if ammonium is present. The presence of chloride, ammonium, nitrate, and bacteriological contamination indicates possible contamination from sewer systems, solid waste, or animal waste [42]. Two samples in the study area exceeded an ammonium concentration of 0.5 mg L^−1^.

The vulnerability of water for human consumption to contamination is a significant concern for the health sector since it is used for hygiene and recreational purposes as well as for consumption. It is necessary to identify the possible sources of contamination and the routes of exposure through which the population ingests the contaminant in order to assess the risk to the human health of a particular population. In Mexico, there exists legislation (NOM-127-SSA1-1994), which specifies the established permissible limits of quality, use, and consumption. This legislation must be respected. The presence of dissolved and insoluble chemicals in the water, which may be of natural or anthropogenic origin, defines the physical and chemical composition of the water.

The study shows that the evaluated parameters comply with the Mexican standard, as they are within the established limit; however, some parameters, such as (NO_3_^−^), (PO_4_^3−^), are outside the norm. According to the results, the water quality found in San Mateo Texcalyacac is physically affected by pollutants, but it is suitable for human consumption. The presence of nitrogen, phosphate, and anthropogenic dissolved organic matter in drinking water represents a risk to human health. Nitrates, chlorides, sulfates, and the chemical activity of phosphorus favor the incorporation of trace elements (heavy metals and metalloids), considered carcinogens. Phosphates and sulfate precipitation with divalent cations (Ca^2+^ and Mg^2+^) promote a deficiency of essential minerals in the groundwater used for human use and consumption. This can cause cardiovascular diseases in those who consume such water.

### 3.2. Geochemical Facies and Groundwater Hydrogeochemistry

A series of diagrams are often used to identify hydrogeochemical processes and perform groundwater quality analysis. Piper diagrams (also known as trilinear diagrams), Mifflin diagrams, and Gibbs graphs were used in the present study as tools for visualizing the relative abundance of common ions in the water samples to obtain the hydrogeochemical facies and have a better understanding of the main processes that control the chemical composition in the groundwater flow system. 

The Piper diagram enables groundwater samples to be classified according to the concentrations of the major ions present in them. The hydrogeochemical facies of groundwater samples are the Mg-HCO_3_ water type, with a tendency (S2 and S3) to the Na-HCO_3_ water type. S1, in the fourth sample, is of the Na-SO_4_-Cl water type, possibly due to the influence of the nearby landfill. The study found that Mg^2+^ is the dominant cation followed by Na^+^, while the anion HCO_3_^−^ was dominant, followed by Cl^−^ and SO_4_^2−^. This is evident in the Piper diagram (Figure 3). The ionic exchange of Na^+^ and K^+^ for Ca^2+^ and Mg^2+^ present on the surface of clay minerals can cause a higher concentration of these ions in the water (r = 0.876, Na^+^ vs. Ca^2+^ and r = 0.980 K^+^ vs. Mg^2+^) (Table 2).

The Gibbs diagram is useful for establishing the relationship between the water composition and the lithological characteristics of the aquifer [43]. The Gibbs diagram presents three regions: the dominance of precipitation, evaporation, and rock-water interaction (Figure 4). The Gibbs diagram shows that the primary process that controls the chemical composition at the first three sampling points is the water-rock interaction of recharged freshwaters with more significant evolution in sampling periods 4 and 5; however, in samples S4 and S5, a slight trend of the meteoric precipitation process is observed, which suggests that there is a temporary effect on the composition of the water.

The Na^+^ + K^+^ vs. Cl^−^ + SO_4_^2−^ graph, proposed by Mifflin [44], allows for the identification of flow systems and the characterization of evolutionary processes of geogenic or anthropogenic origin. The graph is divided into three areas: the area closest to the source is associated with the water’s short residence time or low circulation. In this area, the samples are associated with local recharge. The area between the lines is associated with a long-distance or longer residence time (intermediate flow), and the ion concentrations increase. Finally, in the water-rock interaction zone, the concentration of ions is higher and is associated with a more significant evolution of groundwater (regional flow). According to Figure 4, the groundwater in the area (S1–S3) obeys a local flow with a tendency toward regional flow; while samples S4 and S5 present a similar behavior in all samples (Figure 5d,e); that is, greater evolutionary process and a longer residence time.

Different scatter diagrams were used to identify the hydrogeochemical evolution processes responsible for water quality (Figure 6 and Figure 7). In Figure 5a, the plot of Ca^2+^ + Mg^2+^ versus HCO_3_^−^ + SO_4_^2−^ showed that all water samples are positioned below the line of 1:1 ratio and towards HCO_3_^−^ + SO_4_^2−^, which may indicate that there is a dissolution of carbonate and sulfate minerals, together with a cation exchange process [45]. However, the study area aquifer is in a volcanic environment. There are no sulfate or carbonate minerals in the site geology; consequently, the enrichment of these ions may be associated with the mineralization of organic matter from a nearby solid waste landfill (Figure 6a). Mineralization is the transformation of organic compounds into inorganic ones; during the mineralization of organic carbon present in water, inorganic carbon is released in the form of CO_2_, HCO_3_^−^, CO_3_^−^ and CH_4_, as well as nitrogen, phosphorus, and sulfur in their soluble forms NH_4_^+^, NO_3_^−^, H_2_PO_4_^−^, HPO_4_^2−^ and SO_4_^2−^ [46].

The Na^+^ + K^+^ versus Total cations plot showed that all sample points lie slightly below the line of 1:1 ratio, suggesting that Na^+^ and K^+^ ions are not dominant in the analyzed groundwater samples (Figure 6b). On the contrary, cationic activity such as silicate erodes the precipitation of Ca^2+^ as calcium carbonate and the enrichment of Mg^2+^ in the groundwater. In addition to this, agricultural activity, one of the primary sources of K, is present in the study area, but it is not dominant [46].

According to the results, groundwater can be affected by its proximity to the landfill, as shown in Figure 7, where there is a greater presence of Cl^−^ ions over SO_4_^2−^ ions, with an increase in Cl^−^ ions over time in most samples (Figure 7b). Likewise, the dissolution of gases from the atmosphere, mainly CO_2_ and minerals related to CO_3_^−^, in the unsaturated area and during precipitation and infiltration, gives rise to HCO_3_^−^ type water facies, as observed in the study [47], while the high contribution of Cl^−^ in groundwater is generally due to the dissolution of halite. However, the aquifer under investigation is in a volcanic environment; hence, the abundance of Cl^−^ ions can be associated with incorporating leachate from the sanitary landfill.

Nitrate in groundwater possibly comes from nitrification and denitrification in groundwater near landfills (Figure 7a). Nitrate contributes to the identification of anthropogenic contamination in groundwater. The graphs shown above provide a way to infer the negative effect on the alteration of the quality of groundwater destined for human consumption due to the location near the solid waste disposal sites. Figure 7a confirms that the water in the samples closest to the solid waste (S1–S3) are influenced by this site, while the sample near the Lerma River (S4) receives influence from the river as shown by the inverse correlation between nitrate and sulfate ions (r = −776).

### 3.3. Excitation-Emission Matrix (3D Fluorescence Spectra)

The characterization of dissolved organic matter by 3D fluorescence spectroscopy can be a reliable parameter for identifying or ensuring contamination by incorporating wastewater or leachate from solid waste disposal sites. A 3D fluorescence spectrum provides information on the origin of the organic matter present in the water samples, which can be natural (humic and fulvic acids) (Figure 8c), or anthropogenic (microbiological degradation compounds and aromatic proteins) (Figure 8a,b,d). 

With the 3D fluorescence spectra obtained, the anthropic origin of the dissolved organic matter (DOM) was established using fluorescence measurements [6,7,27,29]. Based on the excitation-emission matrix (EEM) on the relative maximum intensity between each pair of peaks and the presence or absence of DOM, the samples were classified into three types: (1) Type I: sample with two protein peaks (A and B) high-intensity fluorescence (Figure 8a,b); (2) Type II: samples with two humic peaks (C and D) of high fluorescence intensity and presence of nitrogen (Figure 8c); (3) Type III: samples with a protein peak (B) of high fluorescence intensity and two humic peaks (C and D) to its right (Figure 8d).

According to the results obtained, all types of samples indicate anthropogenic contamination in the groundwater. In these samples, the high-intensity protein peaks A and B predominate and; therefore, the intensity ratio between these and the humic peaks for uncontaminated waters 1:3 is not fulfilled [6,7]. The ratio of intensities of the protein peaks and those corresponding to natural organic matter (humic and fulvic) are associated with a possible anthropic origin of said organic matter, since in natural uncontaminated water, the ratio of these peaks is 1:3 [6,7,27,48]. On the other hand, intensity ratios of higher protein peaks could indicate the presence of anthropic contamination, as occurs in samples S1 (ratio = 2.33, 2.35), S2 (ratio = 2.41, 2.68) and S4 (ratio = 2.25). The dissolved organic matter analysis confirmed anthropogenic organic contamination in four of the groundwater samples analyzed (S1, S2, S3, and S4). The fluorescence spectra evidence the presence of microbiological degradation products, aromatic proteins (S1, S2, and S4), natural organic matter, humic and fulvic acids, and nitrate (S3) (Figure 8).

## 4. Conclusions

Groundwater is one of the most important reservoirs for water supply with different uses in the world; in Mexico, there is great research interest in its evolution and behavior. This research indicates that the predominant flow system is local, and the primary process that controls the physicochemical groundwater quality is the water-rock interaction. The parameter values for all water samples were below the permissible limit established by the WHO for drinking water and complied with Mexican regulations. However, the enrichment of sulfate and carbonate ions may be associated with the mineralization of organic matter, and the abundance of PO_4_^3−^, and Cl^−^ ions can be associated with incorporating leachate from the sanitary landfill. Fluorescence spectroscopy analysis showed that the water is polluted with anthropogenic dissolved organic matter; therefore, this technique can investigate groundwater pollution characteristics and monitor DOM dynamics in groundwater. The water quality for human consumption receives input from the solid waste disposal site and the Lerma River. The presence of nitrogen, phosphate, and anthropogenic dissolved organic matter in drinking water represents a risk to human health. The data obtained gives evidence of the influence of landfills and wastewater on the aquifer’s water quality.

## Figures and Tables

**Figure 1 ijerph-18-11195-f001:**
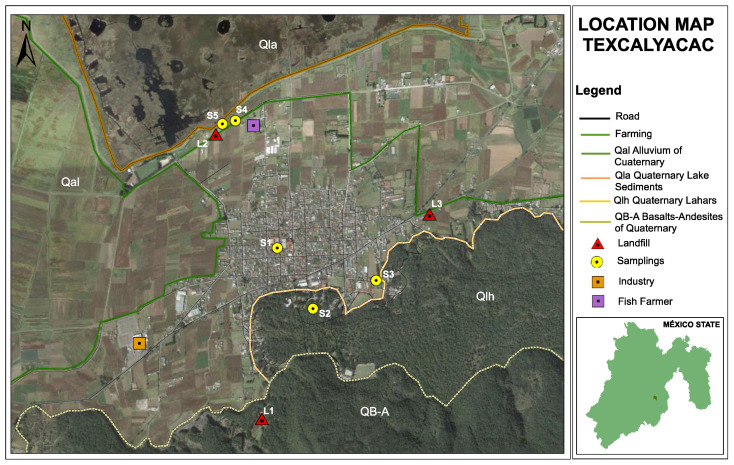
Groundwater sampling locations. Geological context provided.

**Figure 2 ijerph-18-11195-f002:**
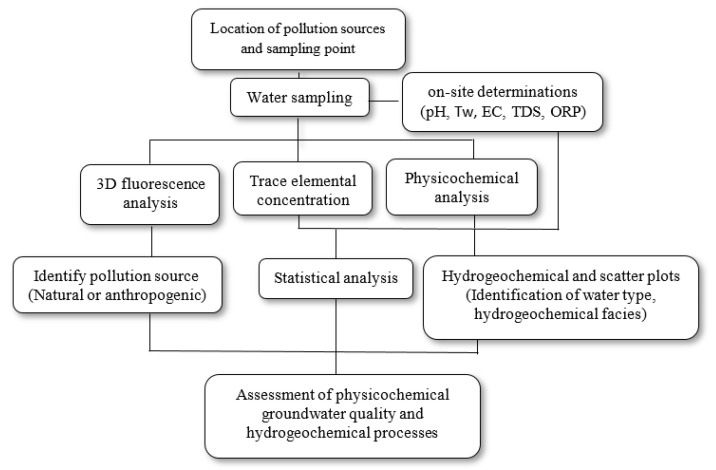
Flow diagram of the methodology used to determine water quality in the study area.

**Figure 3 ijerph-18-11195-f003:**
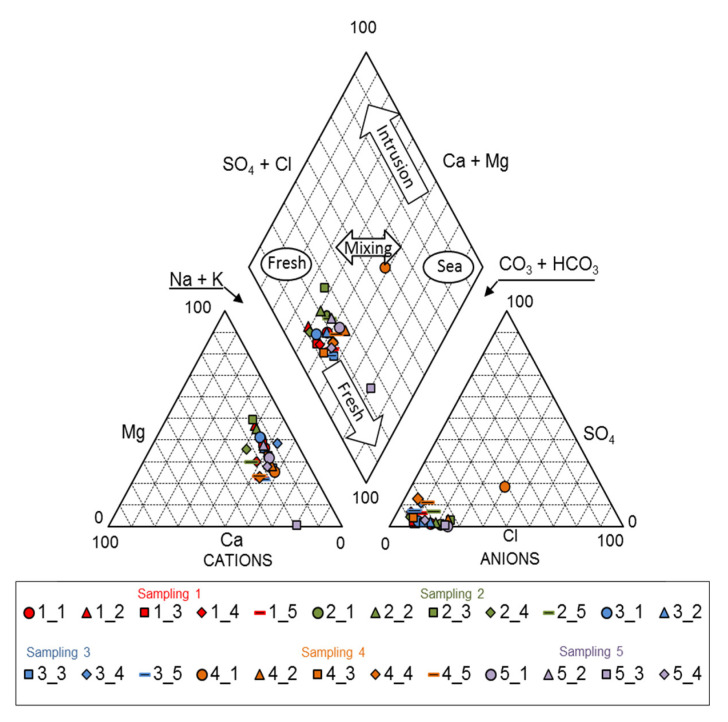
Hydrogeochemistry facies of the groundwater using a Piper diagram.

**Figure 4 ijerph-18-11195-f004:**
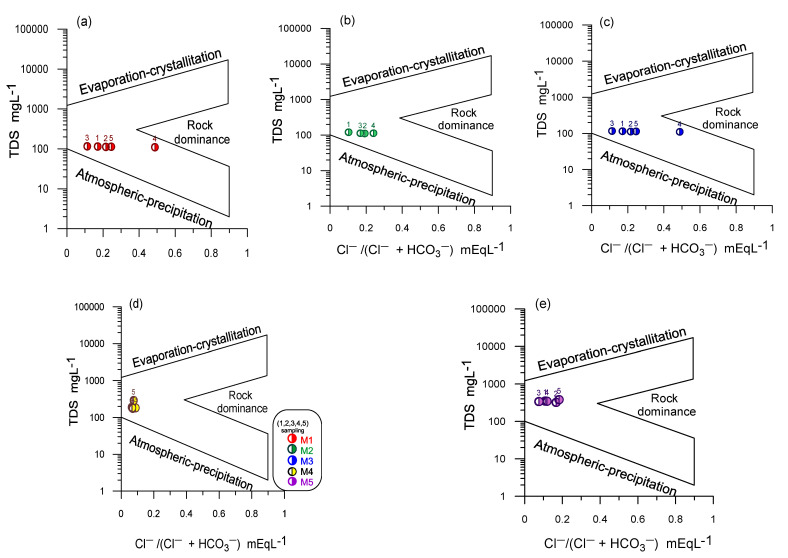
Gibbs diagram, hydrogeochemical processes that predominate in the aquifer and the chemical composition of the water in the study area during all sampling periods: (**a**) S1, (**b**) S2, (**c**) S3, (**d**) S4, and (**e**) S5.

**Figure 5 ijerph-18-11195-f005:**
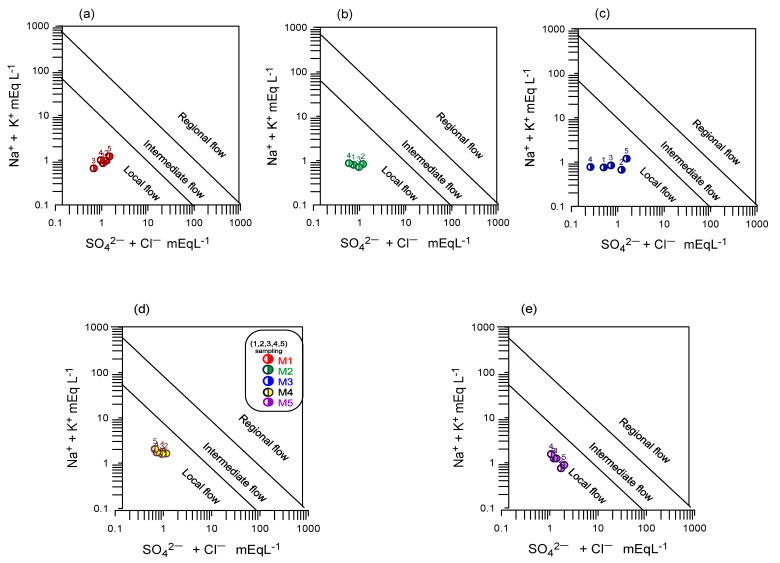
Mifflin diagram, classification of the water according to its evolution during all sampling periods: (**a**) Sample S1, (**b**) Sample S2, (**c**) Sample S3, (**d**) Sample S4, and (**e**) Sample S5.

**Figure 6 ijerph-18-11195-f006:**
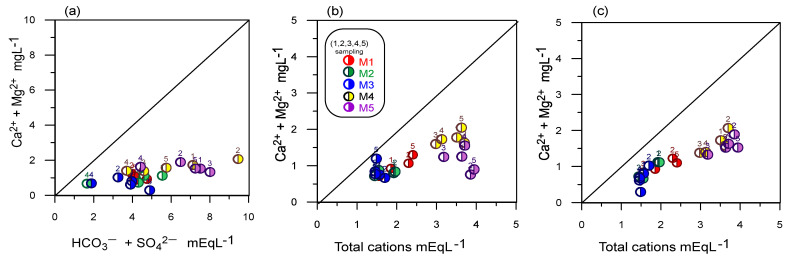
Scatter plot for the hydrogeochemical evolutionary process, (**a**) scatter plot between (Ca^2+^ + Mg^2+^) Vs. (HCO_3_^−^ + SO_4_^2−^), (**b**) scatter plot between (Na^+^+ K^+^) Vs. Total cations, (**c**) scatter plot (Ca^2+^ + Mg^2+^) Vs. Total cations.

**Figure 7 ijerph-18-11195-f007:**
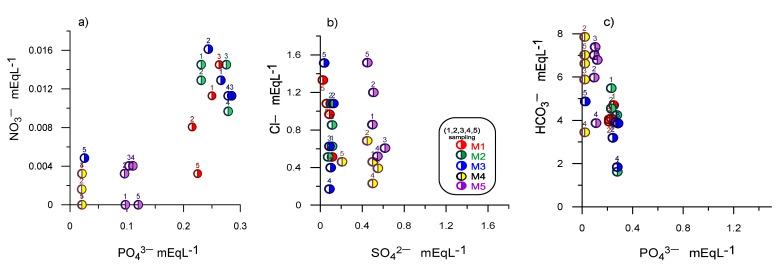
Scatter plot for the hydrogeochemical evolutionary process, (**a**) scatter plot NH_4_^+^ vs. SO_4_^2−^, (**b**) scatter plot Cl^−^ vs. SO_4_^2−^, (**c**) scatter plot HCO_3_^−^ Vs. SO_4_^2−.^

**Figure 8 ijerph-18-11195-f008:**
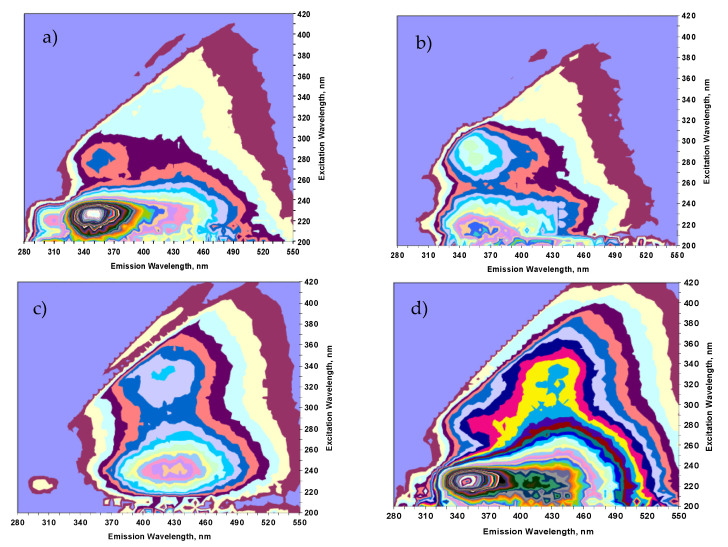
Excitation-Emission Matrix of the groundwater samples from the Municipality of Texcalyacac: (**a**) S1 (Type I), (**b**) S2 (Type I), (**c**) S3 (Type II), (**d**) S4 (Type III).

**Table 1 ijerph-18-11195-t001:** Physicochemical parameters and major constituents of bore-hole water samples taken at Texcalyacac.

Sample	pH	ORP	Tw	EC	TDS	Hard	BOD	COD	DO	PO_4_^3−^	Cl ^−^	N-NO_3_ ^−^	NH_4_^+^	SO_4_^2−^	HCO_3_^−^	Na^+^	Mg^2+^	K^+^	Ca^2+^
			°C	μS/cm	mgL^−^1														
WHO	6.5–8.5	*	25	750	500	300	*	*	5–7	2	250	*	45	250	*	200	50	10	100
NOM-127-SSA1-1994	6.5–8.5	*	*	750	1000	*	*	*	*	*	250	0.5	10	400	*	200	*	*	*
	**sampling 1**																		
S1	6.56	46.60	23.34	227.00	115.00	60.00	0.00	6.55	7.24	7.90	34.35	0.70	0.00	4.20	286.84	19.82	7.90	2.44	5.36
S2	6.44	47.40	23.11	231.00	120.00	80.00	0.00	7.29	7.28	7.30	22.23	0.90	0.00	5.40	334.68	16.56	10.43	3.13	5.08
S3	6.54	47.20	23.16	227.00	112.00	50.00	0.00	11.56	7.14	8.40	14.14	0.80	0.00	4.80	239.04	16.18	6.98	1.97	4.47
S4	6.43	43.79	24.15	379.00	188.00	64.00	0.00	3.66	7.48	0.65	16.35	0.00	0.00	23.98	403.86	36.89	12.14	6.78	14.36
S5	7.78	50.1	21.14	330.00	349.00	169.00	44.86	76.40	7.54	3.2	30.43	0.00	0.535	23.86	427.67	41.40	8.83	12.03	15.77
	**sampling 2**																		
S1	9.16	47.60	21.27	223.00	112.00	160.00	0.00	7.33	7.11	6.80	38.39	0.50	0.00	3.00	240.19	22.42	10.93	3.54	6.37
S2	9.21	48.20	21.08	223.00	111.00	150.00	0.00	8.68	7.25	7.30	38.39	0.80	0.00	4.80	277.67	17.42	10.27	3.10	5.24
S3	9.22	47.90	21.10	224.00	111.0	130.00	0.00	10.35	7.37	7.70	38.39	1.00	0.00	6.00	194.76	13.76	9.68	2.75	4.40
S4	9.15	48.70	21.12	379.00	176.00	160.00	0.00	121.7	6.25	0.53	24.30	0.10	0.00	21.52	549.59	33.20	15.17	7.25	16.09
S5	7.91	50.00	21.30	325.00	323.00	301.00	76.80	320.73	7.57	3.07	42.62	0.20	0.552	24.32	364.52	37.23	12.65	13.26	16.77
	**sampling 3**																		
S1	8.25	46.90	22.16	236.00	116.00	50.00	0.00	156.13	6.04	8.30	18.19	0.90	0.00	5.40	246.68	15.12	7.51	2.09	4.34
S2	8.21	48.00	22.16	229.00	113.00	130.00	0.00	136.81	6.48	8.70	30.31	0.90	0.00	5.40	258.82	15.53	6.16	1.66	4.21
S3	8.21	46.10	23.29	229.00	113.00	100.00	0.00	138.00	6.56	9.00	22.23	0.70	0.00	4.20	235.29	18.56	4.63	1.29	4.60
S4	8.67	64.80	21.12	379.00	182.00	98.00	4.14	21.27	10.17	0.70	13.89	0.20	0.00	26.38	246.11	33.93	8.86	4.74	12.28
S5	7.76	50.3	21.44	293.00	341.00	225.00	11.00	17.54	7.54	3.32	21.54	0.25	0.548	29.62	450.75	37.42	7.69	8.67	13.78
	**sampling 4**																		
S1	8.96	25.92	23.10	230.00	110.00	30.00	20.30	48.74	6.24	8.50	30.31	3.17	0.60	19.02	54.77	23.11	5.37	1.73	5.65
S2	8.95	48.20	22.80	222.00	113.00	40.00	14.20	49.90	6.23	8.80	18.19	0.60	0.10	3.60	98.78	19.31	5.13	1.49	4.83
S3	8.90	49.00	22.90	221.00	112.00	90.00	7.30	50.73	6.20	8.80	6.06	0.70	0.00	4.20	112.10	16.88	5.58	1.53	4.40
S4	8.19	18.35	23.43	379.00	182.00	46.00	12.54	55.11	6.06	0.73	8.18	0.20	0.10	22.88	195.16	36.89	8.25	4.76	14.28
S5	7.72	49.13	21.34	426.00	346.00	114.00	26.36	65.61	6.02	3.54	18.43	0.25	0.548	26.15	236.13	41.91	9.67	9.84	16.48
	**sampling 5**																		
S1	7.78	50.1	21.14	330.00	113.00	190.00	3.40	29.16	7.24	7.10	47.28	0.20	0.10	1.20	250.0	28.06	8.91	3.12	7.21
S2	7.91	50.00	21.30	325.00	115.00	220.00	0.70	26.19	7.33	7.00	51.32	0.50	0.00	3.00	296.0	23.82	10.03	3.31	6.51
S3	7.76	50.3	21.44	293.00	115.00	200.00	3.50	27.78	7.35	9.9	53.75	0.30	0.40	1.80	297.0	27.32	0.11	0.18	5.69
S4	7.72	48.41	22.92	379.00	289.00	221.00	4.83	54.64	6.99	0.71	32.28	0.00	0.10	9.95	337.86	42.43	11.53	7.86	12.32
S5	7.70	48.61	23.52	772.00	382.00	446.00	5.13	64.04	7.09	3.84	53.86	0.00	0.542	21.56	414.38	48.09	9.96	12.67	13.93
max	9.22	64.8	24.15	772.00	382.00	446.00	76.8	320.73	10.17	9.90	53.86	3.17	0.6	29.62	549.59	48.09	15.17	13.26	16.77
min	6.43	18.35	21.08	221.00	110.00	30.00	0.00	3.66	6.02	0.53	6.06	0.00	0.00	1.20	54.77	13.76	0.11	0.18	4.21
aver	8.04	46.86	22.19	308.44	178.36	140.96	9.40	60.64	7.03	5.67	29.02	0.55	0.17	12.29	281.95	27.33	8.57	4.85	8.98
SD	0.88	8.35	1.02	118.27	96.14	95.13	17.60	70.46	0.85	3.24	13.91	0.64	0.24	10.08	113.29	10.61	3.09	3.86	4.85

WHO: World Health Organization; NOM-127-SSA1-2000: Official Mexican Standard for drinking water; *: Not specified in the regultions.

**Table 2 ijerph-18-11195-t002:** Correlation coefficient matrix of groundwater samples of Texcalyacac Municipality.

	DQO	DBO	pH	T	OD	PO_4_^3−^	Alk	SDT	Cl^−^	ORP	NO_3_^−^	Na^+^	Mg^2+^	K^+^	Ca^2+^	SO_4_^2−^	EC	Hrd	NH_4_^+^
DQO	1																		
DBO	0.745 **	1																	
pH	0.378	0.325	1																
T	−0.198	−0.294	−0.493	1															
OD	−0.283	0.155	0.067	−0.332	1														
PO_4_^3−^	−0.044	−0.256	−0.111	0.465	−0.419	1													
Alk	0.151	0.206	−0.007	0.051	−0.143	−0.679 **	1												
SDT	0.329	0.632 *	0.202	−0.095	0.233	−0.692 **	0.585 *	1											
Cl^−^	0.250	0.404	0.516 *	−0.272	−0.149	.319	−0.155	−0.091	1										
ORP	−0.140	0.113	0.312	0.038	0.703 **	−0.399	0.135	0.503	−0.254	1									
NO_3_^−^	−0.119	−0.320	−0.078	0.349	−0.312	0.940 **	−0.671 **	−0.739 **	0.280	−0.325	1								
Na^+^	0.161	0.417	0.104	−0.304	0.345	−0.929 **	0.661 **	0.867 **	−0.254	0.423	−0.971 **	1							
Mg^2+^	0.091	0.299	0.201	−0.601 *	0.039	−0.683 **	0.630 *	0.346	0.194	−0.134	−0.589 *	0.541 *	1						
K^+^	0.446	0.768 **	0.271	−0.340	0.204	−0.753 **	0.666 **	0.909 **	0.106	0.294	−0.778 **	0.854 **	0.637 *	1					
Ca^2+^	0.296	0.522 *	0.208	−0.370	0.320	−0.929 **	0.770 **	0.813 **	−0.144	0.390	−0.925 **	0.935 **	0.661 **	0.905 **	1				
SO_4_^2−^	0.187	0.437	0.195	−0.226	0.491	−0.823 **	0.668 **	0.761 **	−0.242	0.630 *	−0.776 **	0.813 **	0.407	0.774 **	0.905 **	1			
EC	0.370	0.685 **	0.252	−0.067	0.275	−0.650 **	0.604*	0.971 **	−0.036	0.558*	−0.694 **	0.814 **	0.312	0.909 **	0.825 **	0.839 **	1		
Hrd	0.477	0.707 **	0.612 *	−0.305	0.043	−0.422	0.391	0.778 **	0.429	0.286	−0.481	0.593 *	0.421	0.797 **	0.590 *	0.435	0.747 **	1	
NH_4_	0.453	0.775 **	0.306	0.091	0.130	−0.310	0.416	0.862 **	0.208	0.461	−0.394	0.546 *	0.127	0.791 **	0.572 *	0.610 *	0.918 **	0.776 **	1

** The correlation is significant at the 0.01 level; * The correlation is significant at the 0.05 level.

## Data Availability

The study did not report any data.

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
