# Peer review of "Assessment of Physicochemical Groundwater Quality and Hydrogeochemical Processes in an Area near a Municipal Landfill Site: A Case Study of the Toluca Valley"

_ijerph, 2021, doi:10.3390/ijerph182111195_

Round 1
Reviewer 1 Report
The study is interesting and addresses the importance of knowing the quality of the aquifer, since the water can be used for human consumption. However, there are some observations to make.
It would be interesting to know the characteristics of the landfill, since it has not been described at any time (type of waterproofing, waste stored, distance to the samples taken, how leachates are managed, distance to the water table ... It is important to know this information in order to be able to establish the influence or not of the landfill on the contamination of the aquifer Have leachate samples been analyzed to compare with the aquifer samples?
In the same way, it would be interesting to locate industries, crops and activities that could also cause contamination of the aquifer, since the area seems to be anthropized. It is assumed that it is the landfill that pollutes, but it could be any other activity present.
Please put the units of population density (pag 2. Study area)
Which means "A case study of the Toluca Valley." at the end of page 2? I think that sentence has "moved" from place.
In figure 1, it would be interesting to name each sample (S1, S2 ...)
What are the dates (month and year) of sampling?
There are some citations (like page 4. Kiros et al.) That are not well referenced (with numbers). Please review the entire manuscript.
Table 1: Some of the values shown as "maximum" and "minimum" are not found in the sample values. For example, the maximum pH value is 9.32, which sample does it correspond to?
The maximums of EC and DO also do not coincide with any sample. And some mean values are not well calculated (like NH4 +). This table needs to be thoroughly reviewed to locate errors. I hope you have registered and saved the correct values, so as not to distort the test.
It is very curious that the sulfate value in all the S4 samples is exactly the same (23.98). It's right? Why did this happen? What is the location of sample S4 on the map? I think this deserves to be commented. It is assumed that the samples have been taken in different months to consider the dry and wet seasons, right?
The same is true for calcium and magnesium (and more parameters) in that same sample. You have to review all this well.
Please be consistent with the number of decimal places throughout the table.
Good and bad written subscripts and superscripts alternate throughout the manuscript. Please write them all well, with the correct spelling.
In table 2, the anions and cations are not spelled correctly (the corresponding plus and minus signs are missing)
Despite all of the above, the research is interesting and with a little extra work it can be published
Author Response
Dear reviewer,
The authors appreciate the observations made to improve the document, every one of them was addressed.
1.- It would be interesting to know the characteristics of the landfill, since it has not been described at any time (type of waterproofing, waste stored, distance to the samples taken, how leachates are managed, distance to the water table ... It is important to know this information in order to be able to establish the influence or not of the landfill on the contamination of the aquifer Have leachate samples been analyzed to compare with the aquifer samples?
The authors added the following paragraph
The final disposal of solid waste is defined as permanently depositing waste in sites and facilities, whose characteristics prevent its release into the environment, and the consequent effects on the population and ecosystems should not be significant (Gómez et al., 2013). A landfill is a site assigned by a municipality to deposit its solid waste until it is time to be transferred to its final disposal. Therefore, these are temporary and must comply with the characteristics assigned by Civil Protection; however, some landfills exceed the lifetime and continue to function without a specific regulation and control, so by not having control, the Leachates begin to infiltrate the aquifer until they encounter the groundwater and thus begin the contamination process. Actuality in San Mateo Texcalyacac ​, there is a landfill that exceeds its life span and does not have management and regulation of its solid waste, having a negative impact on the physical environment and human health since the site is open-air and there is no leachate control.
2.- In the same way, it would be interesting to locate industries, crops and activities that could also cause contamination of the aquifer, since the area seems to be anthropized. It is assumed that it is the landfill that pollutes, but it could be any other activity present.
The requested information was added to the map.
3.- Please put the units of population density (pag 2. Study area)
The change was made
4.- Which means "A case study of the Toluca Valley." at the end of page 2? I think that sentence has "moved" from place.
The change was made
5.- In figure 1, it would be interesting to name each sample (S1, S2 ...)
The change was made
6.- What are the dates (month and year) of sampling?
The change was made
7.- There are some citations (like page 4. Kiros et al.) That are not well referenced (with numbers). Please review the entire manuscript.
The change was made
8.- Table 1: Some of the values shown as "maximum" and "minimum" are not found in the sample values. For example, the maximum pH value is 9.32, which sample does it correspond to?
The change was made
9.- The maximums of EC and DO also do not coincide with any sample. And some mean values are not well calculated (like NH4 +). This table needs to be thoroughly reviewed to locate errors. I hope you have registered and saved the correct values, so as not to distort the test.
The change was made
10.-It is very curious that the sulfate value in all the S4 samples is exactly the same (23.98). It's right? Why did this happen? What is the location of sample S4 on the map? I think this deserves to be commented. It is assumed that the samples have been taken in different months to consider the dry and wet seasons, right?
The change was made
11.- The same is true for calcium and magnesium (and more parameters) in that same sample. You have to review all this well.
The change was made
12.- Please be consistent with the number of decimal places throughout the table.
The change was made
13.- Good and bad written subscripts and superscripts alternate throughout the manuscript. Please write them all well, with the correct spelling.
The change was made
14.- In table 2, the anions and cations are not spelled correctly (the corresponding plus and minus signs are missing)
The change was made

Reviewer 2 Report
Reading the title and having a first glance at the abstract it looked a promising paper, as it addresses major issue of the sustainability assessment of a city that has received extensive scientific interest lately, the quality assessment of areas near municipal landfill site. I carefully read the article. The topic is highly relevant and fit well into the scope of the “Environmental research and Public Health” Journal. The conceptualization article is very interesting for the readers of the journal, but a number of significant improvements have to be made before published. In this light, I suggest a major revision of the article in order to merit publication. I hope that the criticisms I present below in bullet form will help the author improve the paper.
My main objections for publishing the paper in its current form are the following:
- I would expect much more elaboration on the introductory section. Within this section, the authors should provide clear and detailed analysis of the basic scope of the paper. In its current form the introductory section is rather short and general and does not provide any scientific originality. The authors should definitely add the aim of the paper.
- The authors should give a more detailed analysis and clarifications on the methodological advances of the presented approach. I am not completely convinced about the innovation of the methodological approach adopted in the paper.
- Section 2 ”Materials and Method” is rather short and does not provide added value in the manuscript. Εven though the issue addressed by the manuscript is very important the methodology section (presented in section 2) proposed is not convincing. I would suggest to the authors to restructure and reorganize methodology in order to be more reader-friendly. In this light, I would suggest the creation of a descriptive flowchart depicting the major steps of the proposed methodology. The exact flow of the methodology framework that have been utilized within the context of this study is not clear.
- I would expect more discussion within the justification of the overview of the results.
- In the conclusion’s section, I would expect some more managerial insights and general comments, rather than a repetition of study results. The authors should clearly reconstruct this section.
- As a final comment, I should repeat that the paper needs further explanations. I expect more discussion, more interpretation, based on the above comments. In principal the paper should convince that it does make methodological advances. In conclusion, even though the issue addressed by the manuscript is very important the methodology proposed is not convincing without addressing first the points highlighted above.
Author Response
Dear reviewer,
The authors appreciate the observations made to improve the document, every one of them was addressed.
The authors reviewed, rearranged, and supplemented the introduction section.I would expect much more elaboration on the introductory section. Within this section, the authors should provide clear and detailed analysis of the basic scope of the paper. In its current form the introductory section is rather short and general and does not provide any scientific originality. The authors should definitely add the aim of the paper.
In the text the objective was added
Therefore, the present study aimed to evaluate groundwater quality and identify the hydrogeochemical processes in the aquifer near an unregulated municipal landfill.
The authors should give a more detailed analysis and clarifications on the methodological advances of the presented approach. I am not completely convinced about the innovation of the methodological approach adopted in the paper.
Section 2 ”Materials and Method” is rather short and does not provide added value in the manuscript. Εven though the issue addressed by the manuscript is very important the methodology section (presented in section 2) proposed is not convincing. I would suggest to the authors to restructure and reorganize methodology in order to be more reader-friendly. In this light, I would suggest the creation of a descriptive flowchart depicting the major steps of the proposed methodology. The exact flow of the methodology framework that have been utilized within the context of this study is not clear.
The authors revised, reorganized, and completed this section; the suggested diagram was elaborated.
I would expect more discussion within the justification of the overview of the results.
In the conclusion’s section, I would expect some more managerial insights and general comments, rather than a repetition of study results. The authors should clearly reconstruct this section.
The conclusions section was modified

Round 2
Reviewer 1 Report
Dear authors,
Thank you for considering my comments. I think that this way the manuscript is more refined and presents a good investigation.
Author Response
Dear reviewer,
The authors appreciate the observations made to improve the document, every one of them was addressed.
We added one more paragraph to complete the information in the introduction section, and conclusions have been revised and improved accordingly.
We appreciate the time and dedication in reviewing the manuscript.

Reviewer 2 Report
The authors have adopted all the proposed revisions, as such the manuscript have been modified accordingly
Author Response

(The authors gave the same response as above.)
